# Intracellular Hydrolysis of Small-Molecule *O*-Linked *N*-Acetylglucosamine Transferase Inhibitors Differs among Cells and Is Not Required for Its Inhibition

**DOI:** 10.3390/molecules25153381

**Published:** 2020-07-25

**Authors:** Elena Maria Loi, Matjaž Weiss, Stane Pajk, Martina Gobec, Tihomir Tomašič, Roland J. Pieters, Marko Anderluh

**Affiliations:** 1Faculty of Pharmacy, University of Ljubljana, 1000 Ljubljana, Slovenia; Elena.Maria.Loi@ffa.uni-lj.si (E.M.L.); Matjaz.Weiss@ffa.uni-lj.si (M.W.); Stane.Pajk@ffa.uni-lj.si (S.P.); Martina.Gobec@ffa.uni-lj.si (M.G.); Tihomir.Tomasic@ffa.uni-lj.si (T.T.); 2Department of Chemical Biology & Drug Discovery, Utrecht Institute for Pharmaceutical Sciences, Utrecht University, P.O. Box 80082, NL-3508 TB Utrecht, The Netherlands; R.J.Pieters@uu.nl

**Keywords:** ester hydrolysis, inhibitor, *O*-GlcNAc transferase, OGT inhibitor

## Abstract

*O*-GlcNAcylation is an essential post-translational modification that occurs on nuclear and cytoplasmic proteins, regulating their function in response to cellular stress and altered nutrient availability. *O*-GlcNAc transferase (OGT) is the enzyme that catalyzes this reaction and represents a potential therapeutic target, whose biological role is still not fully understood. To support this research field, a series of cell-permeable, low-nanomolar OGT inhibitors were recently reported. In this study, we resynthesized the most potent OGT inhibitor of the library, OSMI-4, and we used it to investigate OGT inhibition in different human cell lines. The compound features an ethyl ester moiety that is supposed to be cleaved by carboxylesterases to generate its active metabolite. Our LC-HRMS analysis of the cell lysates shows that this is not always the case and that, even in the cell lines where hydrolysis does not occur, OGT activity is inhibited.

## 1. Introduction

*O*-GlcNAc transferase (OGT) is a ubiquitous cellular enzyme responsible for one of the most important post-translational protein modifications: the transfer of GlcNAc moiety to serine or threonine residues [1]. The *O*-GlcNAcylation cell status is controlled by the availability of the cofactor UDP-GlcNAc and the expression of OGT and OGA (*O*-GlcNAcase), the latter catalyzing the opposite reaction of *O*-GlcNAcylation, namely the cleavage of *O*-glycosidic bond between GlcNAc and the protein. As UDP-GlcNAc level depends mostly on the intracellular concentration of glucose that fuels hexosamine biosynthetic pathway, OGT activity reflects the cell nutritional status and influences many critical cellular processes, including lipid droplet remodeling, mitochondrial functioning, epigenetic control of gene expression, and proteostasis [2]. Furthermore, it shares the same substrates as kinases and, therefore, interferes with kinase-dependent signaling [3,4]. However, in contrast to numerous kinases, *O*-GlcNAcylation is catalyzed by one enzyme only: OGT, which exists in three isoforms differing in the tetratricopeptide region that influences the protein substrate specificity [5].

Despite the evident importance of *O*-GlcNAcylation in shaping cellular protein activity, the phenotypic cell response upon OGT modulation is still not entirely understood. This is partly because of the lack of proper molecular tools in the past, e.g., selective and potent molecular probes that would permeate the cell membrane and potently inhibit OGT. Although a number of small molecule OGT inhibitors were reported in the literature [6,7,8,9,10,11], not many fulfil the above-mentioned requirements, with 2-acetamido-1,3,4,6-tetra-*O*-acetyl-2-deoxy-5-thio-α-d-glucopyranose or Ac-5SGlcNAc being one of the rare suitable probes [12]. Just very recently, a series of potent OGT inhibitors based on a previous hit (OSMI-1) were synthesized and evaluated in the Walker laboratory (Harvard University, Boston, MA, USA) [13]. These inhibitors (Figure 1) distinguish themselves by nanomolar K_d_ values on recombinant OGT and low micromolar inhibition of global *O*-GlcNAcylation in HEK293T cell lines. All these compounds were prepared in the ester form and the corresponding free acids, and the authors postulated that upon cell permeation, intracellular esterases cleave esters rapidly, so that the corresponding carboxylates are the active species in cells. In our quest to find a suitable phenotypic assay for potential OGT inhibitors and to study in-depth cancer and immune cellular response following OGT inhibition, we have studied the effect of one of these OGT inhibitors (OSMI-4 in its free acid and ester forms) in selected cell lines (K562, HEK293 and HEK293T). As these compounds can easily undergo intramolecular cyclization during synthesis yielding the diketopiperazine derivative (OSMI-4 DKP, Figure 2), we first aimed to rule out that this could occur in the cellular environment. Intriguingly, we found that, despite the presence of intracellular esterases, in some cell lines, the compound remains in the ester form throughout the whole experiment. Furthermore, as we have observed cell viability inhibition in those cases, we concluded that the ester form is also an OGT inhibitor in cells, besides free carboxylate. In fact, the ester was even found to be a slightly more potent inhibitor of recombinant enzyme than the parent carboxylate. This conclusion may have a significant impact on our understanding of the OGT inhibitors’ mechanism of action and their future design.

## 2. Results and Discussion

OSMI-4b and its corresponding carboxylic acid (OSMI-4a) were obtained according to the original synthetic protocol [13] with minor modifications aimed at improving the final yield (Appendix A: longer reaction times for steps *a_2_* and *e*, Boc deprotection of **3** by HCl instead of TFA). During the synthesis, we observed the rapid formation of the corresponding diketopiperazine, OSMI-4 DKP, from the ester, presumably by the attack of the deprotonated sulfonamide to the ester carbonyl group (Appendix A); hence, we hypothesized that the same intramolecular reaction could occur in the cellular environment and could influence OGT inhibition and the subsequent effects.

To prove it and to use the diketopiperazine as a standard for cell lysate analyses, we have synthesized it by activating the carboxylic acid (OSMI-4a) in alkaline conditions and performing the subsequent cyclization assisted by a coupling reagent (Appendix A). In particular, our docking experiments suggested that the cyclization would lock the inhibitor in an unfavorable conformation, impairing the binding to OGT.

To study the effect of the OGT inhibitor on *O*-GlcNAcylation, we selected three different cell lines: chronic myelogenous leukemia cells (K562), embryonic kidney cells (HEK293), and their variant (HEK293T). After treatment with various concentrations of OSMI-4a and OSMI-4b (10–80 µM), the metabolic activities of the cells were assessed in an CellTiter 96 Aqueous One Solution Cell Proliferation (MTS) Assay (Figure 3) and the levels of intracellular *O*-GlcNAcylation by Western blot analysis (Figure 4). A well-known OGT inhibitor of the same family, OSMI-1, was used as a control in the experiments [8].

Upon direct administration of OSMI-4a, we could not observe any significant changes in metabolic activity or global *O*-GlcNAcylation, which was expected as the free carboxylate should not permeate the cell membrane easily. Conversely, treatment with the ethyl ester led to a decrease of global *O*-GlcNAcylation in all the cell lines, which was in line with the assumption of Walker et al.

Hence, we proceeded to verify which of the possible metabolites was responsible for the effect. The LC-MS chromatogram of the isolated compounds provided their retention times and accurate masses (Table 1, Appendix A), allowing us to develop a standard method to confirm their presence in complex mixtures. Afterwards, the samples of the cell lysates were submitted to LC-HRMS analysis (Appendix A). In cells treated with carboxylate OSMI-4a, only traces of inhibitor could be detected in the LC-HRMS spectra (Appendix A). Therefore, the fact that we could not observe OGT inhibition in any cell lines can be attributed to the lack of permeability of the free carboxylates, as mentioned before. Moreover, the presence of the diketopiperazine OSMI-4 DKP was never observed in the cell lysates after the administration of the ester OSMI-4b, suggesting that the intramolecular cyclization was not occurring in the cellular environment. Interestingly, when the cells were treated with OSMI-4b, even though OGT inhibition could be clearly observed in all three cell lines, the ester metabolism was not always the same. In K562 cell lysates, signals for both ester OSMI-4b and acid OSMI-4a were observed 5 and 72 h after treatment, as expected (Appendix A). In both embryonic kidney cell lines, only the presence of the ester was confirmed even after 72 h after cell treatment (Appendix A). This could be explained by the fact that HEK cells are characterized by a lower expression of intracellular esterases [14]. However, in both HEK cell lines, notably lower *O*-GlcNAcylation levels were observed. As only the ester OSMI-4b was detected in these cell lines, contrary to initial beliefs, we can conclude that that ester itself is responsible for OGT inhibition and that diminished *O*-GlcNAcylation levels are a direct consequence of ester inhibitory activity.

To corroborate this result, we decided to measure the inhibitory activity of all assayed compounds on the recombinant enzyme. For this purpose, we selected two different assays: the UDP-Glo™ Assay (Promega, Fitchburg, WI, USA), which measures UDP formation in glycosyltransferase reactions, and a direct fluorescent activity assay recently developed by Vocadlo’s group [15]. Surprisingly, not only could all three compounds OSMI-4a, OSMI-4b, and OSMI-4 DKP efficiently inhibit OGT activity, but both assays proved that the ethyl ester OSMI-4b was the most potent inhibitor of the series, with slightly lower IC_50_ values in both assays than for free acid OSMI-4a (Table 2). This indisputably shows that, in cells with lower esterase expression, OGT inhibition, and, consequently, the *O*-GlcNAcylation status, depends solely on the ester activity. This observation is consistent with the crystal structure of OSMI-4a in the OGT active site (PDB ID: 6MA1), which clearly shows that the carboxylate of the inhibitor does not exhibit any significant interaction with OGT, as it points towards the solvent (Figure 5). Accordingly, the ester in the same position should not be detrimental to binding or should even contribute to binding, since its potential lower partial desolvation penalty upon binding would cause lower enthalpic cost than for the corresponding carboxylate. By using the molecular docking of OSMI-4 DKP into the OGT active site, we have shown how the potential formation of the diketopiperazine would lead to a loss of potency; the binding pose of OSMI-4 DKP and its comparison with the crystal structure of OSMI-4a clearly show that the diketopiperazine would bind to OGT in a suboptimal conformation (Figure 5).

Based on the above, we speculate that for the future development of promising OSMI-4-based OGT inhibitors (Figure 2), it could be beneficial to replace the ethyl ester with a more stable functional group (e.g., amide, *t*-butyl ester). This change could potentially increase the potency, stability, and drug properties of the inhibitors in the cell lines where the carboxylesterases are largely expressed and omit the need for an ester as a prodrug form for in vivo studies. Even if the ethyl ester form would be administered in vivo, liver or plasma esterases would probably cleave it before reaching the target cells. On the contrary, a stable amide or other replacement for an ester would allow passive absorption and would reach the target tissues due to metabolic stability.

## 3. Materials and Methods

### 3.1. Cell Culture

HEK-293 (ATCC) and HEK-293T (ATCC) cell lines were cultured in Dulbecco’s Modified Eagle Medium (Sigma-Aldrich, St. Louis, MO, USA). K562 cell line (ATCC) was cultured in Roswell Park Memorial Institute 1640 medium (RPMI-1640) (Sigma-Aldrich, MO, USA). Both mediums were enriched with 10% fetal bovine serum (Gibco, Grand Island, NY, USA), 2 mM L-glutamine, 100 U/mL penicillin, and 100 μg/mL streptomycin (all from Sigma-Aldrich, MO, USA). Cells were cultured in a humid atmosphere at 37 °C and 5% CO_2_.

### 3.2. Metabolic Activity Assay

After the cells were seeded into 96-well plates at 8000 cells/mL (100 μL/well), they were treated with various concentration of OSMI-4a, OSMI-4b, OSMI-1, or the corresponding vehicle, as a control. After 96 h treatment, the metabolic activity was assessed using the CellTiter 96 Aqueous One Solution Cell Proliferation Assay (Promega, WI, USA). The absorbance was measured at 492 nm on an automated microplate reader BioTek Synergy HT (BioTek Instruments, Inc., Norwich, UK). The results are presented as the percentage of metabolic activity of the control cells stimulated with the vehicle (mean + SD) from two to three independent experiments, each performed in duplicate.

### 3.3. Western Blot Analysis

HEK293, Hek293T, and K562 cells were seeded in 6-well culture plates at a concentration of 1 × 10^6^ cells/mL and treated with the compound of interest or corresponding vehicle for 24 h. After the indicated time point, cells were harvested, washed in ice-cold PBS, and lysed in RIPA buffer (50 mM Tris-HCl, pH 7.4, 150 mM NaCl, 1% NP-40, 0.5% Na-deoxycholate, 1 mM EDTA) with 1× Halt Protease and Halt Phosphatase Inhibitor Cocktail (Thermo Scientific, Pierce Biotechnology, Rockford, IL, USA). The lysates were sonicated, rocked on ice for 30 min, and centrifuged at 15,000× *g* at 4 °C for 15 min. Supernatants containing 20 μg of protein were heated at 96 °C for 5 min in a sample loading buffer (3% SDS, 10% glycerol, 62.5 mM Tris-HCl, pH 6.8, 5% 2-mercaptoethanol and 0.1% bromphenol blue). Protein samples were electrophoresed in 8% SDS-polyacrylamide gels and then transferred to nitrocellulose membranes (GE Healthcare Life Science, Chicago, IL, USA) by wet electroblotting. Nonspecific binding sites were blocked for 1 h at room temperature in Tris-buffered saline (TBS)-Tween (0.1%) containing 3% bovine serum albumin (Sigma-Aldrich, St Louis, USA). The membranes were then washed and incubated overnight at 4 °C with gentle stirring with appropriate dilutions of primary antibodies in (TBS)-Tween (0.1%). The primary antibodies were anti-*O*-GlcNAcylation (BioLegend, San Diego, CA, USA) diluted 1:1000 and anti-ß-actin (Sigma-Aldrich, St Louis, MO, USA) diluted 1:7000. Following incubation with the primary antibody, membranes were washed three times and incubated for 1 h at room temperature with the corresponding dilution of the appropriate secondary antibody conjugated with horseradish peroxidase (Cell Signaling Technology, Leiden, The Netherlands). The immunoreactivity of respective proteins of interest was determined by chemiluminescence using the SuperSignal West Femto substrate (ThermoScientific, Pierce Biotechnology, IL, USA), in accordance with the manufacturer’s instructions. To ensure the equal loading of proteins, the membranes were stripped and reprobed with appropriate antibodies under the same conditions as those described above.

### 3.4. Statistical Analysis

GraphPadPrism software (v 8.2.1) was used for statistical analysis. ANOVA with Dunnett’s multiple comparison test was used for comparisons between treated samples and control sample, to detect statistical significance between individual pairs. *p* < 0.05 was considered statistically significant.

### 3.5. Cell Permeability

Cells were seeded in T-75 culture plates at a concentration of 1 × 10^6^ cells/mL and treated with the compound of interest or corresponding vehicle for 5 or 72 h. After the indicated time points, cells were harvested, washed two times in ice-cold PBS, resuspended in 150 µL of water/methanol (4:1) mixture, and stored at −20 °C overnight. Cells were sonicated, rocked on ice for 30 min, and centrifuged at 15,000× *g* at 4 °C for 15 min. Supernatants were transferred to new tubes and treated with 750 µL of methanol and stored at −20 °C overnight. The next day, samples were centrifuged at 15,000× *g* at 4 °C for 15 min, and supernatants were transferred to new tubes. Samples were dried in nitrogen atmosphere at 40 °C for approximately 1 h and dissolved in 150 µL of 20% methanol/water or in 50% Acetonitrile/water mixture.

Samples were analyzed by LC-MS system, which included Thermo Scientific UltiMate 3000 UHPLC liquid chromatograph and Thermo Scientific Exactive Plus Hybrid Quadrupole-Orbitrap mass spectrometer. Chromatographic separation was performed on Waters Acquity UPLC BEH C18 column (50 × 2.1 mm, 1.7 µm particles), kept at 45 °C. The injection volume was 1.00 µL. The compounds were separated using mobile phase A consisting of water–acetonitrile–formic acid (99:1:0.1, *v*/*v* ratio) and mobile phase B consisting of water–acetonitrile–formic acid (1:99:0.1, *v*/*v* ratio). The flow rate was 0.30 mL/min with the following gradient: 0–12.0 min, 5%–95% B; 12.0–17.0 min, 95% B; 17.0–18.0 min, 95%–5% B; 18–21 min 5% B. The mass spectrometer was operated in HESI positive mode with the following MS parameters: sheath gas flow rate, 25 (arbitrary units); auxiliary gas flow rate, 10 (arbitrary units); capillary temperature, 350 °C; and spray voltage, 3.5 kV. Mass analysis was performed only between 5.3 min and 7.7 min after injection, since during this interval all compounds of interest eluted.

### 3.6. UDP-Glo™ Assay

This assay evaluates *O*-GlcNAcylation through monitoring UDP formation in glycosyltransferase reactions by luminescence. Briefly, OGT reactions were carried out in a 50-μL final volume, containing 0.1 mM UDP-GlcNAc, 200 nM purified full-length OGT, 100 μM RBL-2 peptide in OGT reaction buffer (25 mM Tris-HCl, pH 7.5; 1 mM DTT; 12.5 mM MgCl_2_). Reactions were incubated at 37 °C for 2 h. Afterwards, each reaction was transferred in duplicate into a 96-well white microplate and was mixed with a 1:1 ratio of the UDP-Glo Detection Reagent. After incubation at room temperature for 1 h, the luminescence was recorded with a POLARstar^®^ Omega microplate reader (BMG LABTECH) or with a BioTek Synergy™ H4 microplate reader. The data were plotted with GraphPad prism software, version 8, [Inhibitor] vs. response-variable slope.

### 3.7. Fluorescent Activity Assay

The fluorescent activity assay was performed as recently published [15]. OGT reactions were carried out in a 25-μL final volume, containing 2.8 μM glycosyl donor BFL-UDP-GlcNAc, 1.6 μM purified full-length OGT, 9.2 μM glycosyl acceptor HCF-1 Serine in OGT reaction buffer (1 × PBS pH 7.4, 1 mM DTT, 12.5 mM MgCl_2_). Reactions were incubated at room temperature for 1 h, in the presence of different concentrations of inhibitor (the inhibitors were preincubated with OGT for at least 5 min). The reactions were then stopped by the addition of UDP at a final concentration of 2 mM, followed by Nanolink magnetic streptavidin beads (3 μL). After incubation at room temperature for 30 min, the beads were immobilized on a magnetic surface and washed thoroughly with PBS-tween 0.01%. Finally, the beads were resuspended in PBS-tween 0.01% and transferred to a microplate for endpoint fluorescence measurement. Fluorescence was read at Ex/Em 485/530 with a POLARstar^®^ Omega microplate reader (BMG LABTECH). The data were plotted with GraphPad prism software, version 8, [Inhibitor] vs. response-variable slope.

### 3.8. Molecular Docking

For docking with FRED software (OEDOCKING 3.3.1.2: OpenEye Scientific Software, Santa Fe, NM, USA, http://www.eyesopen.com) [16,17,18], the OGT binding site (PDB entry: 6MA1) was prepared using MAKE RECEPTOR (Release 3.3.1.2, OpenEye Scientific Software, Inc., Santa Fe, NM, USA; www.eyesopen.com). The grid box around the ligand OSMI-4a bound in the OGT crystal structure was generated automatically and was not adjusted. This resulted in a box with the following dimensions: 16.00 Å ×21.00 Å × 18.00 Å and the volume of 6048 Å^3^. For “Cavity detection” slow and effective “Molecular” method was used for detection of binding sites. Inner and outer contours of the grid box were also calculated automatically using “Balanced” settings for “Site Shape Potential” calculation. The inner contours were disabled. Ala896 was defined as hydrogen bond donor and acceptor constraint for the docking calculations. The ligands were prepared by OMEGA (Release 3.3.1.2, OpenEye Scientific Software, Inc., Santa Fe, NM, USA; www.eyesopen.com). OSMI-4a was used as a control. The ligands were then docked to the prepared binding site of OGT using FRED (default settings). The results were visualized and analyzed with VIDA (version 4.3.0.4, OpenEye Scientific Software, Inc., Santa Fe, NM, USA, www.eyesopen.com).

## 4. Conclusions

To summarize, we analyzed the presence of OSMI-4b and its derivatives in lysates of different human cell lines, using HPLC coupled to High-Resolution Mass Spectrometry. The formation of diketopiperazine derivative was never observed in vitro, while both carboxylic acid and ester forms were present in the lysates of chronic myelogenous leukemia cells (K562). Interestingly, in human embryonic kidney cells (HEK293 and HEK293T), where esterases are poorly expressed and the ester OSMI-4b remains intact, the effect on cell metabolic activity and global *O*-GlcNAcylation could still be observed, suggesting that the ester itself can inhibit OGT. By using two different assays, we have confirmed that OSMI-4b inhibits the recombinant enzyme potently (UDP-Glo™ Assay and fluorescent activity assay); namely, the ester and free acid had comparable potencies with ester being even slightly more potent.

Our results provide new inputs on the intracellular metabolism of OSMI-4b and its mechanism of action, and we hope that they will support its use in cellular assays and future optimization of OSMI-4-based OGT inhibitors.

## Figures and Tables

**Figure 1 molecules-25-03381-f001:**
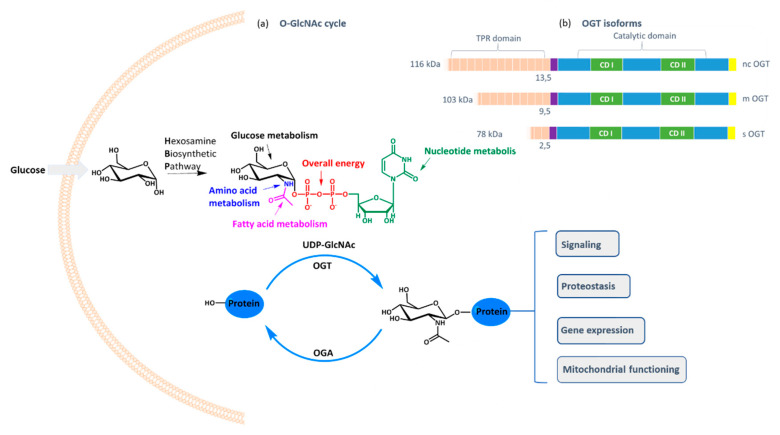
(**a**) Schematic representation of *O*-GlcNAcylation cycle. Approximately 2–3% of the cellular glucose enters the hexosamine biosynthetic pathway (HBP), leading to the formation of UDP-GlcNAc. OGT uses the latter as a glycosyl donor to glycosylate serine and threonine residues of hundreds of proteins, while OGA removes the GlcNAc moiety. OGT activity reflects the cell nutritional status and influences many critical cellular processes. (**b**) Schematic structure of OGT isoforms. The three splice isoforms of OGT, ncOGT, mOGT and sOGT, possess identical catalytic region but differ in the number of tetratricopeptide repeats (TPR) at the *N*-terminus.

**Figure 2 molecules-25-03381-f002:**
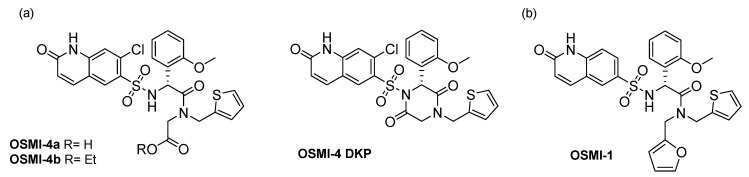
(**a**) Structure of OSMI-4a (carboxylic acid), OSMI-4b (ethyl ester), and the corresponding diketopiperazine derivative, OSMI-4 DKP; (**b**) structure of OGT inhibitor OSMI-1.

**Figure 3 molecules-25-03381-f003:**
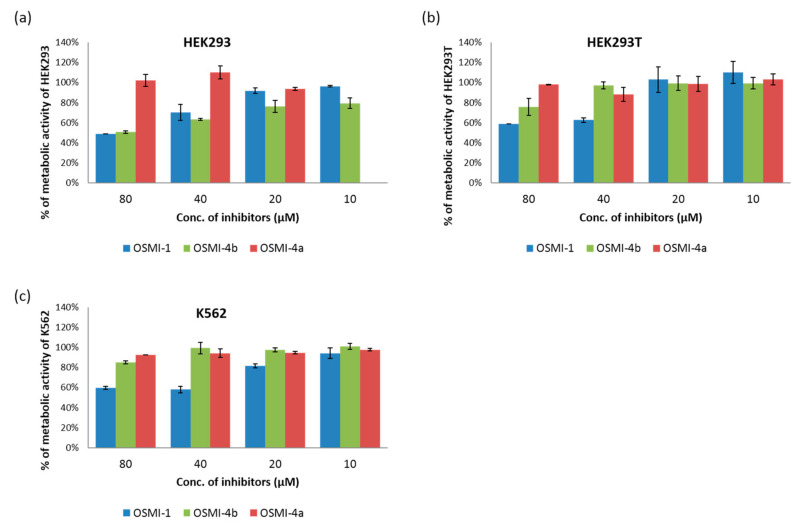
Effects of metabolic activity after 96 h treating HEK293 (**a**), HEK293T (**b**), and K562 (**c**) cells with OSMI-1, OSMI-4a, and OSMI-4b. The results are presented as the percentage of metabolic activity of the control cells stimulated with the vehicle (mean + SD) from two to three independent experiments, each performed in duplicate.

**Figure 4 molecules-25-03381-f004:**
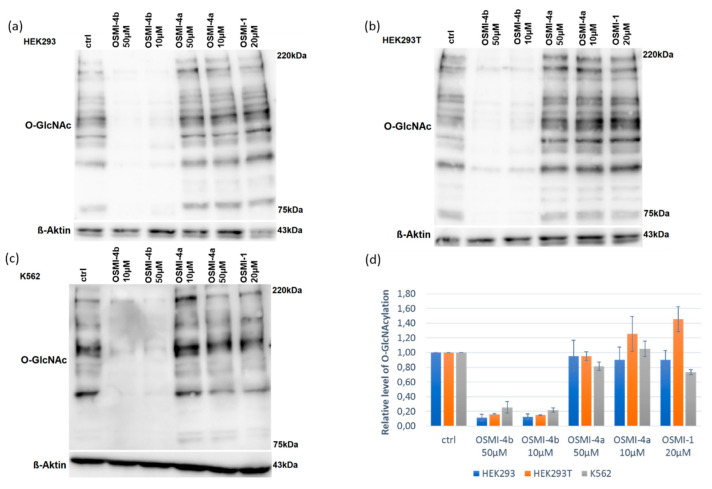
Effects of *O*-GlcNAcylation levels after 24h treating HEK293 (**a**), HEK293T (**b**), and K562 (**c**) cells with OSMI-1, OSMI-4a, and OSMI-4b. Western blotting for *O*-GlcNAc levels after treatment of cells with inhibitors at 10 μM (OSMI-4a, OSMI-4b), 20 μM (OSMI-1), or 50 μM (OSMI-4a, OSMI-4b) show that compound OSMI-4b and OSMI-1 are more effective than OSMI-4a; “ctrl” denotes DMSO control. Dose-dependent decreases in global *O*-GlcNAc levels were observed upon treatment with OSMI-4b in cells HEK293T and K562. (**d**) The western blot results are presented as the relative level of *O*-GlcNAcylation of the control cells stimulated with the vehicle from two to three independent experiments.

**Figure 5 molecules-25-03381-f005:**
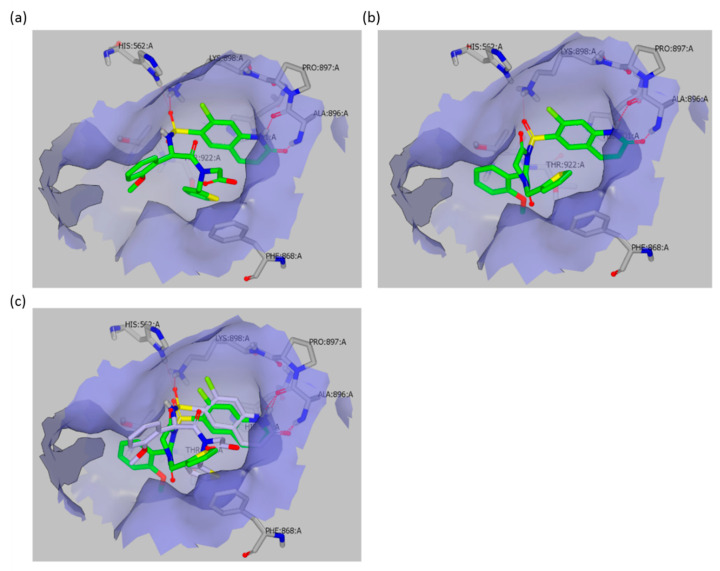
(**a**) Representation of the binding pose of OSMI-4a in OGT active site (PDB: 6MA1 10.2210/pdb6MA1/pdb); (**b**) predicted binding mode of OSMI-4 DKP in OGT active site. Docking was performed using FRED algorithm on OEDocking software (OEDOCKING 3.3.0.2: OpenEye Scientific Software, Santa Fe, NM, USA. http://www.eyesopen.com) (Materials and Methods); (**c**) representation of OSMI-4 DKP (green) and OSMI-4a (grey) overlapped in OGT binding pocket. Pictures prepared with Vida (VIDA, version 4.3.0.4, OpenEye Scientific Software, Inc., Santa Fe, NM, USA, www.eyesopen.com).

**Table 1 molecules-25-03381-t001:** High resolution UHPLC–PDA–Q-Orbitrap identification of compounds OSMI-4a, OSMI-4b, and OSMI-4 DKP.

Peak	Retention Time (min)	Elemental Composition [M + H]^+^	Theoretical Mass (*m*/*z*)	Measured Mass (*m*/*z*)	MS^n^ Ions δppm
**OSMI-4a**	5.77	C_25_H_23_O_7_N_3_ClS_2_	576.0660	576.0655	−0.96
**OSMI-4 DKP**	6.69	C_25_H_21_O_6_N_3_ClS_2_	558.0555	558.0553	−0.33
**OSMI-4b**	7.10	C_27_H_27_O_7_N_3_ClS_2_	604.0973	604.0961	−2.01

**Table 2 molecules-25-03381-t002:** IC_50_ of the three OGT inhibitors measured with UDP-Glo™ Assay (Promega) and fluorescent activity assay. Results are expressed as mean ± SD.

Name	IC_50_ UDP-Glo Assay (µM)	IC_50_ Fluorescent Activity Assay (µM)
**OSMI-4b**	0.5 ± 0.5	0.06 ± 0.02
**OSMI-4a**	1.5 ± 0.6	0.3 ± 0.1
**OSMI-4 DKP**	9 ± 2	5 ± 0.8

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
