# Peer review of "Intracellular Hydrolysis of Small-Molecule O-Linked N-Acetylglucosamine Transferase Inhibitors Differs among Cells and Is Not Required for Its Inhibition"

_molecules, 2020, doi:10.3390/molecules25153381_

Round 1
Reviewer 1 Report
I think the manuscript has potential, but it must be improved significantly before publication. The main claim of the authors is that a recently published compound has OGT inhibitory capacity not only in the acid form but also (or even more so in ester form (which originally was designed to only serve as a way to penetrate the cells). Some of my concerns are below (not in the order of importance):
1. Abstract: "Our LC-HRMS analysis of the cell lysates shows that this is not always the case and that 20 even in the cell lines where hydrolysis does not occur, the effect on OGT activity is preserved." - I think this should read 'inhibited' not preserved.
2. Figure 2. would require more labelling (e.g.: a, b c). Repeating the western blot experiments would be also a must, a single data per condition is not enough. I also have some reservation about the quality of the WBs shown here, there is a lot of background artefact.
3. Too many data are put in the supplementary file. I suggest to revise all the supplementary data, add more informative labelings and more comprehensive figure legends and move them in the main text. E.g. it is not clear which chromatogram corresponds to the cell lysate and which to the starting mixture.
4. Throughout, the number of repeats are either inadequate or not mentioned et all. Scienitific findings can only be falsified if N is known and statistical analysis is performed.
5. The name of the compounds should be used in the same way throughout. Either 1,2, 4 or OSMI-1, OSMI-2, etc. I personally would prefer the abbreviations, not the numbers.
6. Line 98 -101 demonstrates the need of revising the supplementary data. "In cells treated with carboxylate 2, only traces of inhibitor could be detected in the LC-HRMS spectra. Therefore, the fact that we could not observe OGT inhibition in any cell lines can be attributed to the lack of permeability of the free carboxylates, as mentioned before." Which figure supports this notion?
Author Response
I think the manuscript has potential, but it must be improved significantly before publication. The main claim of the authors is that a recently published compound has OGT inhibitory capacity not only in the acid form but also (or even more so in ester form (which originally was designed to only serve as a way to penetrate the cells). Some of my concerns are below (not in the order of importance):
- Abstract: "Our LC-HRMS analysis of the cell lysates shows that this is not always the case and that 20 even in the cell lines where hydrolysis does not occur, the effect on OGT activity is preserved." - I think this should read 'inhibited' not preserved.
As a response to the reviewer's comment, we have modified the sentence with "Our LC-HRMS analysis of the cell lysates shows that this is not always the case and that even in the cell lines where hydrolysis does not occur OGT activity is inhibited."
- Figure 2. would require more labelling (e.g.: a, b c). Repeating the western blot experiments would be also a must, a single data per condition is not enough. I also have some reservation about the quality of the WBs shown here, there is a lot of background artefact.
We agree that the quality of the Western blot experiment was modest, so we repeated the western blot experiments, and provided good quality images with lower background artefact (in Figure 4). Indeed, the additional labelling allows an easier examination of the data presented in the figure.
- Too many data are put in the supplementary file. I suggest to revise all the supplementary data, add more informative labelings and more comprehensive figure legends and move them in the main text. E.g. it is not clear which chromatogram corresponds to the cell lysate and which to the starting mixture.
As a response to the reviewer's comment, figure S2 and figure S13 were moved in the main text; they are now named figure 3 and figure 5, respectively. Figure 3 presents additional description in the caption.
In the SI, the LC-MS chromatograms are now labelled in the picture as well for more clarity.
- Throughout, the number of repeats are either inadequate or not mentioned et all. Scienitific findings can only be falsified if N is known and statistical analysis is performed.
As a response to the reviewer's comment, we repeated the Western blot experiments (see point 2) and we peformed additional the IC50 measurement using UDP-Glo assay (figure S9). The new experiments just corroborated previously obtained results, and we agree that we can obtain the same claims with more confidence.
- The name of the compounds should be used in the same way throughout. Either 1,2, 4 or OSMI-1, OSMI-2, etc. I personally would prefer the abbreviations, not the numbers.
As a response to the reviewer's comment, we changed the name of the compounds to abbreviations instead of numbers. The three compounds are now named as follows: OSMI-4a (carboxylic acid), OSMI-4b (ethyl ester), OSMI-4 DKP (diketopiperazine).
- Line 98 -101 demonstrates the need of revising the supplementary data. "In cells treated with carboxylate 2, only traces of inhibitor could be detected in the LC-HRMS spectra. Therefore, the fact that we could not observe OGT inhibition in any cell lines can be attributed to the lack of permeability of the free carboxylates, as mentioned before." Which figure supports this notion?
As a response to the reviewer's comment, references to the specific figures in SI were added throughout the main text.
Reviewer 2 Report
Molecules 831154
Intracellular Hydrolysis of Small-Molecule O-linked 2 N-acetylglucosamine transferase Inhibitors Differs 3 Among Cells and Is Not Required for its Inhibition.
The manuscript is well structured and presented; it is a good manuscript.. Some minor suggestions are given below, after which the manuscript should be accepted for publication.
- If possible, Insert a figure 1 showing the process described in the preceding lines could be added after the line37, pag1.
- Lines 45-46, page 2
Walker laboratory (please insert City and Country).
- Lines 66-669. Page 2
Please insert reference bibliographic, which support the paragraph.
- Please insert between lines 98-99 an additional table with Table
High resolution UHPLC–PDA–Q-Orbitrap identification of compounds 1,2 and 3
|
Peak |
Retention time ( min) |
UV |
Elemental composition [M + H]+ |
Theoretical mass (m/z) |
Measured mass (m/z) |
MSn ions δppm |
|
1 |
|
|
|
|
|
|
|
2 |
|
|
|
|
|
|
|
3 |
|
|
|
|
|
|
- Please change in supplementary Figure .S3
exctracted by extracted.
Author Response
Intracellular Hydrolysis of Small-Molecule O-linked 2 N-acetylglucosamine transferase Inhibitors Differs 3 Among Cells and Is Not Required for its Inhibition.
The manuscript is well structured and presented; it is a good manuscript. Some minor suggestions are given below, after which the manuscript should be accepted for publication.
- If possible, Insert a figure 1 showing the process described in the preceding lines could be added after the line37, pag1.
As a response to the reviewer's comment, the Figure 1 was inserted to schematically represent the O-GlcNAc cycle described in the preceding lines.
- Lines 45-46, page 2
Walker laboratory (please insert City and Country).
As a response to the reviewer's comment, we specified the City and Country (Harvard University, Boston, United States) where the Suzanne Walker laboratory is located.
- Lines 66-69. Page 2
Please insert reference bibliographic, which support the paragraph.
We inserted the reference to the paper that includes the original synthetic protocol (Martin et al. Structure-Based Evolution of Low Nanomolar O-GlcNAc Transferase Inhibitors. Journal of the American Chemical Society 2018, 140, 13542–13545, doi:10.1021/jacs.8b07328).
- Please insert between lines 98-99 an additional table with Table
High resolution UHPLC–PDA–Q-Orbitrap identification of compounds 1,2 and 3
|
Peak |
Retention time ( min) |
UV |
Elemental composition [M + H]+ |
Theoretical mass (m/z) |
Measured mass (m/z) |
MSn ions δppm |
|
1 |
|
|
|
|
|
|
|
2 |
|
|
|
|
|
|
|
3 |
|
|
|
|
|
|
We inserted a table using the suggested template.
- Please change in supplementary Figure .S3 exctracted by extracted.
We corrected the misspelling of the word extracted.
Round 2
Reviewer 1 Report
Most of my previous concerns were met by the authors. Few minor corrections that I still recommend:
- the font size of some of the figures are really small, and if one zooms in, the resolution becames blurry. E.g. the western-blot images, molecular weight texts are hard to read.
- I appreciate that the experiments were replicated and error bars were shown now. However statistical analysis of the data and discussing the significances are still missing. Please also include the description of the statistical analysis in the materials and methods.
- Why are the cell line names supplemented with _B (underscoreB)?
- Please include +/-SD values for Table 2.
- It is not completely clear what the authors mean by TIC and EIC (total and extracted ion chromatogram), especially for the cell lysates? Is this before and after ethyl-acetate extraction? Please clarify how you prepare the cell lysate samples before analysis. This might explain what I still don't understand about this claim: "only traces of inhibitor could be detected in the LC-HRMS spectra (Figure S7)." Which chromatogram supports this? The TIC chromatograms show a peak at 5.84 min. - this is not OSMI-4a, right? But the EIC chromatogram (which I suppose is prepared from the TIC sample) clearly shows a peak at the right time; 5.77 min.
Author Response
In the following lines you may find our point-to-point reply to Reviewer #1 comments.
Most of my previous concerns were met by the authors. Few minor corrections that I still recommend:
- The font size of some of the figures are really small, and if one zooms in, the resolution becames blurry. E.g. the western-blot images, molecular weight texts are hard to read.
As a response to reviewer comment, we increased the font size of figure 1, figure 3 and figure 4.
- I appreciate that the experiments were replicated and error bars were shown now. However statistical analysis of the data and discussing the significances are still missing. Please also include the description of the statistical analysis in the materials and methods.
As a response to reviewer comment, we included the description of the statistical analysis in the material and methods.
- Why are the cell line names supplemented with _B (underscoreB)?
"_B" referred to one of the replicates of the western-blot experiment and it was removed to avoid confusion.
- Please include +/-SD values for Table 2.
As a response to reviewer comment, we included +/-SD values in Table 2.
- It is not completely clear what the authors mean by TIC and EIC (total and extracted ion chromatogram), especially for the cell lysates? Is this before and after ethyl-acetate extraction? Please clarify how you prepare the cell lysate samples before analysis.This might explain what I still don't understand about this claim: "only traces of inhibitor could be detected in the LC-HRMS spectra (Figure S7)." Which chromatogram supports this? The TIC chromatograms show a peak at 5.84 min. - this is not OSMI-4a, right? But the EIC chromatogram (which I suppose is prepared from the TIC sample) clearly shows a peak at the right time; 5.77 min."
The protocol used to prepare the cell lysate samples is reported in materials and methods, in the "cell permeability" section, while the description for the TIC and EIC chromatograms is the following:
-TIC – abbreviation for total ion chromatogram; it is created by summing up intensities of all mass spectral peaks belonging to the same scan. TIC includes background noise as well as sample components.
-EIC – abbreviation for extracted ion chromatogram; one or more m/z values representing one or more analytes of interest are recovered ('extracted') from the entire data set for a chromatographic run.
In the TIC of most cell lysates, there is a lot of background and signals for compounds of interest are not visible (a notable exception is seen in Fig. S3 a and c, signal for ester at 7.11 min). Nevertheless, the presence of ester or corresponding acid in cell lysates can clearly be seen in EIC.
The peak at 5.84 min present in the TIC does not correspond to OSMI-4a.
It is true that signals at 5.77 min in EIC (Fig. S7) correspond to the acid. However, the signal is extremely weak, supported by small NL (intensity of base peak, normalized level) value in the upper right corner of each chromatogram and the noise present in the peak itself; therefore only a very small amount of compound entered the cell. Chromatograms are however normalized to the highest signal even if the peak is minute in absolute terms.